# *Stop Uploading Test Data in Plain Text*: Practical Strategies for Mitigating Data Contamination by Evaluation Benchmarks

**Alon Jacovi**[1]    **Avi Caciularu**[1,2]    **Omer Goldman**[1]    **Yoav Goldberg**[1,3]

[1] Bar Ilan University
[2] Google Research
[3] Allen Institute for Artificial Intelligence
alonjacovi@gmail.com

## Abstract

Data contamination has become prevalent and challenging with the rise of models pretrained on large automatically-crawled corpora. For closed models, the training data becomes a trade secret, and even for open models, it is not trivial to detect contamination. Strategies such as leaderboards with hidden answers, or using test data which is guaranteed to be unseen, are expensive and become fragile with time. Assuming that all relevant actors value clean test data and will cooperate to mitigate data contamination, what can be done? We propose three strategies that can make a difference: (1) Test data made public should be encrypted with a public key and licensed to disallow derivative distribution; (2) demand training exclusion controls from closed API holders, and protect your test data by refusing to evaluate without them; (3) avoid data which appears with its solution on the internet, and release the web-page context of internet-derived data along with the data. These strategies are practical and can be effective in preventing data contamination.

## 1 Introduction

Common NLP models today are large language models trained on data crawled from the internet (Raffel et al., 2020; Bommasani et al., 2022; Touvron et al., 2023). This data is often malformed or obfuscated in ways that make it difficult to audit at scale (Bommasani et al., 2022; Mitchell et al., 2023). In particular, *evaluation data* that is also available on the web may be used as part of training, and it can be challenging to verify whether it was used in practice or not (OpenAI, 2023; Google, 2023a).[1] Worse, for many closed models, training data is considered a trade secret and thus unknown to the research community. Such models are being evaluated on data that cannot be certified to be

unseen during training (Brown et al., 2020). Indeed, signs show that such models were exposed to test data (Dodge et al., 2021; Magar and Schwartz, 2022), and we refer to this as *data contamination*.

The above issue of internet-crawled training data is one of two prominent scenarios of data contamination we consider in this work. The second is in the access to closed models via APIs.[2] Such models are frequently used in research for various purposes (Wei et al., 2023; Qin et al., 2023; Moor et al., 2023), and thus they are also evaluated (Srivastava et al., 2022; Bubeck et al., 2023). In most cases, the institution behind the API reserves the option to use the data sent to them as training data in further iterations.[3] In this case, any valuable evaluation data sent to closed API models for any purpose is potentially compromised for any subsequent evaluations using the same data.

The NLP evaluation community is now witness to *two urgent crises*: Data contamination in training data crawled from the internet, and in training data collected from calls to a closed API. The implications are severe—not only is much of our evaluation methodology potentially compromised, but we also *cannot fully identify* the scope and magnitude of the contamination, even for open models.

We engage with the two crises by outlining three practical strategies that individual researchers can enact to protect the integrity of their evaluations:

→ **Strategy 1:** Protect data from automatic crawlers using public key encryption and a license that forbids distribution of adaptations ("no derivatives").

---

[1]For example, OpenAI (2023) found that the BIG-Bench benchmark (Srivastava et al., 2022) was compromised to an extent that prevented its usage entirely.

[2]E.g., OpenAI's GPT series (Brown et al., 2020), MosaicML Inference (MosaicML, 2023), and Google's Bard and PaLM API (Google, 2023b).

[3]OpenAI currently provides exclusion guarantees for certain API calls as of March 1, 2023. This guarantee does not extend to data sent before this date or to data sent through the ChatGPT and DALL-E Labs interfaces. Google's Bard provides no exclusion guarantee exists as of this writing. Sources: openai.com/policies/api-data-usage-policies; bard.google.com/faq

→ **Strategy 2:** Withhold from evaluating APIs that give no training exclusion options.

→ **Strategy 3:** Avoid data that appears with its solution on the internet. If the data originates from the internet, release its context with it.

## 2 Setting

We consider two independent scenarios of data contamination, and three assumptions underlying our mitigation strategies:

**Scenario 1 (*internet crawled corpora*):** The model to be evaluated is based on a training set derived automatically from the internet. The training set is closed or large enough that it is difficult to exhaustively and routinely search for all possible instances of data contamination from public test sets (both exact and approximate matches).

Note that we include cases in which the model potentially trained on some form of the data's solution, *even if it was not trained on the data verbatim*. For example, in the task of sentiment classification of Amazon product reviews (Johnson and Zhang, 2014), the answer (the review's rating) is available on the internet, and possibly compromised in training, even if the dataset was processed and its final form was not available for the model.

**Scenario 2 (*closed API models*):** The model to be evaluated is a closed model behind an API, and there is no global or conditional guarantee of exclusion from future training. Any API call which contains test data compromises that test data for all models under the API holder and their affiliates.

**Assumption 1 (*presumption of contamination*):** In all cases, if it is *possible* that the training data has been contaminated, we *assume* that it is. In other words, if some test data is accessible to automatic internet crawling systems, we consider it compromised. If it is possible that the API holder is using test data from API calls in training, we consider it compromised. The strategies in this work are designed to protect the evaluation under this strict condition.

**Assumption 2 (*sincere actors*):** We are concerned with a setting where the evaluators and model developers are *non-adversarial*—all actors appreciate clean test data and will not seek to "cheat" evaluations, as they share an incentive to reliably prove the value of their work.

The challenging scenarios in Section 2 primarily stem from a lack of resources and conflicting

business interests rather than adversarial sabotage. Therefore, we assume that all actors are sincere in their desire to keep evaluation data outside of model training, and leave research on strategies against adversarial actors to others.

**Assumption 3 (*contamination inheritance*):** Models that were trained on data derived from other models (Jiao et al., 2020; Taori et al., 2023; Chiang et al., 2023; Hsieh et al., 2023, inter alia), models that are using the weights of other models (Sun et al., 2020; Ganesh et al., 2021; Choshen et al., 2022), and ensembles of other models (Wortsman et al., 2022a,b) will all be considered as contaminated as their "ancestors". This applies even when the ancestor was used to train only a part of the model (e.g., Gao et al., 2023).

## 3 Why is Data Contamination Prevalent?

### 3.1 Closed models give no guarantees or controls about held-out data

Models with private training data—whether the models themselves are closed or not—make it impossible to know if they were trained on particular evaluation data.[4]

Even when using test data that is guaranteed to be unknown to a closed API model (e.g., by using data created after the model was last updated or by using data before it is publicly released), on the *first moment* that this data is used to evaluate the closed model, it *ceases to be unknown*, and there is currently no standardized training exclusion controls by API holders. Furthermore, for much of evaluation research that evaluates models before the data is publicly released, the API holders *will not know* whether the data being sent to them belongs to a would-be evaluation set or not.

### 3.2 Even with open data, detection is difficult

Even for models whose training data is known, it can be challenging to understand exactly what they were trained on or to filter out test data before training (Mitchell et al., 2023).[5] The scale of the corpora and the rapid pace of model development and evaluation development hinders thorough checks. Additionally, repurposing data from the

---

[4]There is research on methods for deriving whether particular data existed in a model's training from the model alone (Carlini et al., 2021, 2023a; Chang et al., 2023). However, such methods are fragile to false negatives.

[5]Some efforts exist to provide tools for auditing training data: E.g., Marone and Durme (2023); C4-Search (2023); Piktus et al. (2023).

internet is common when constructing evaluation data—some of which revoke exact-match or ngram-match detection. Fuzzy matching is expensive to run at scale and must be performed routinely between every evaluation set and pretraining dataset.

Although exhaustive checks are rare in practice, even when performed, *false negatives* and *false positives* are still possible. As noted by OpenAI (2023), who ran a partial exact match, false negatives will manifest when a slightly modified example is in the training data. This issue persists for fuzzy matching, which works based on assumptions (Ukkonen, 1985) and only covers specific cases (Myers, 1999). Data contamination that is manifested in any way that stealthily evades a particular fuzzy-match method will not be found. This is an inherent limitation of data contamination detection, and such cases occur in practice (Razeghi et al., 2022; Carlini et al., 2023b).

Finally, while detection is possible if done exhaustively and routinely—this is a *reactive* measure, not a *preventative* measure, for evaluators who have no control over the training data but full control of the evaluation data. The data which was compromised is to be discarded and replaced. The strategies we propose in this work, and in particular Strategy 1, are preventative.

### 3.3 Existing mitigation strategies are imperfect

There are two strategies to mitigate data contamination in current practice:

**Live leaderboards with hidden answers.** The answers to test data are kept hidden behind an interface that only reveals the evaluation result. There are weaknesses with this approach:

*Partial mitigation*: The test data itself, sans answers, is still available on the internet. This data can be automatically crawled, the hidden answers are at risk of being compromised if the test data is repurposed (Scenario 1) or independently labeled by the model developer behind closed doors without knowledge that it is benchmark data (Scenario 2). And finally, live leaderboards are traditionally only applied to test sets, but *development sets* need protection from training to be effective, as well.

*Rarely enforced*: Live leaderboards are costly and time-consuming to maintain. In practice, this constraint proves to be significantly restrictive: They are rarely implemented, rarely used, and get

discontinued with time.[6]

*Responsibility of evaluation falls to the benchmark host*: In the case of automatic evaluation metrics, this poses no issue, but when evaluation is costly or time consuming—such as with human evaluation—the leaderboard host alone often must shoulder the costs of all evaluations (Khashabi et al., 2022).

**Creating (very) new data.** Most trivially, it is possible to run the evaluation before any data is publicized. Additionally, models sometimes admit some guarantees about the last possible date in which their parameters were updated (Touvron et al., 2023), or otherwise they can be assumed to be frozen within some reasonable leeway (e.g., a minute or an hour). Using data which was only created after the model was last frozen is a strategy to guarantee that the data is unseen, even if the data is public (Liu et al., 2023). Creating counterfactual versions (or other augmentations) of older data achieves the same purpose.

Of course, this strategy is extremely inefficient: *The guarantee vanishes for newer models* so that the evaluation data soon loses relevance. This requires evaluation research to continue to create new data for every evaluation, in an expensive cat-and-mouse game.

## 4 Suggested Mitigation Strategies

### 4.1 *Strategy 1*: Encrypt test data with a public key, and use a "No Derivatives" license

*Conditions: Scenario 1.*

This strategy is simple and cheap yet is an impressively potent guard against non-adversarial crawling of plain text test data in training corpora: *Simply upload the test data after encrypting its contents, alongside the key used to decrypt it.* A simple method is by compressing the data in a password-protected archive (e.g., with zip).

A license with a *No Derivatives* clause, such as "CC BY-ND 4.0",[7] will protect the data from being redistributed without its encryption, while still being otherwise permissive.[8]

---

[6]We randomly selected five datasets with leaderboards from the Allen Institute for Artificial Intelligence's leaderboard website. We found that an average of 4.2 models cited the datasets of these leaderboards, even though they did not submit their results to each corresponding leaderboard, opting to use their development sets instead.

[7]creativecommons.org/licenses/by-nd/4.0/deed

[8]See Section 5 for additional discussion on licenses.

Unlike the strategy of contamination detection, this strategy is *preventative* when employed by evaluation practitioners since it stops the evaluation data from being compromised, to begin with.

Importantly, the goal here is not to protect the data from adversarial actors (as in the case with live leaderboards), but to protect the data from automatic crawling. Therefore, the encryption key can be safely released with the encrypted data, and the encryption method can be simple and fast, so long as it sufficiently distorts the text. However, *we warn against using standard obfuscation or compression methods that are not key-protected*, since some crawling systems include pipelines of automatic decompression or deobfuscation.

Finally, *online dataset hubs* (e.g., Hugging Face Datasets, Lhoest et al., 2021; ThoughtSource, Ott et al., 2023), should *withhold from including test data in their online viewer directly*, and specify when or whether hidden data was previously available in plain text.[9] For showcasing the data, a sample of compromised representative examples can be presented instead.[10]

***Strategy 1 Corollary*: Few-shot prompts.** Few-shot prompts are demonstrations used as training data. Although they are not regarded as evaluation data, they are used primarily to evaluate model generalization under strict data constraints. Few-shot evaluation relies on the assumption that this small number of demonstrations will approximate model behavior on any small number of demonstrations.

Few-shot prompts in the literature are commonly displayed in papers, in online repositories, and in online demos—in plain text. They appear alongside other prompts, other data, and other relevant information that should be considered unintended for their original purpose. These prompts are often reused many times in subsequent work, and thus appear many times on the internet in "unnatural" contexts more-so related to NLP research (e.g., the prompts by Wei et al., 2022). We consider such prompts as compromised. When a model is given

an evaluation prompt which is compromised, the evaluation ceases to be representative.

Therefore, we should consider *prompts with data as evaluation data*: Avoid uploading them to the internet in plain text (including inside papers). Since such prompts are relatively inexpensive to annotate, we should *avoid re-using them at all* when we suspect that they were compromised, and *annotate new prompts* instead.

## 4.2 *Strategy 2*: Refuse to send test data to closed API models until exclusion controls are implemented

*Conditions: Scenario 2.*

Scenario 2 is a strict scenario and difficult to guard against without cooperation from the API host. Since we consider the API host as a sincere actor that values reliable evaluation, there are incentives in place for evaluation practitioners to demand this cooperation and for the API hosts to comply.

Foremost, since the very first API usage during evaluation compromises the test data, this strategy calls for not evaluating the closed API model until this situation changes in order to protect the integrity of the data. This, in turn, pressures the API host to provide appropriate training exclusion controls in order to participate in evaluation practice and research. Mechanically, the API host may comply by implementing a system to request exclusion from future training.[11]

As an *intermediate strategy*, in the absence of an exclusion guarantee, it is possible to prioritize collecting cheaper, "single-use" data that can be used for the purpose of a less-reliable evaluation estimate: (1) By holding out a portion of training data, if it exists; (2) By creating synthetic variants of the data, or synthetic data altogether.

## 4.3 *Strategy 3*: Avoid data that appears with its solution on the internet

*Conditions: Scenario 1.*

In order to reduce data collection costs, data for training and evaluation is commonly repurposed from the internet.[12] The data labels are then either derived automatically from context (e.g., review score for product reviews, Rajeev and Rekha,

---

[9]In the case of Hugging Face Datasets, it is possible to use "*gated repositories*" to block test data from crawler access, although they are not currently used for this purpose. Gating mechanisms and web-page metadata that requests not to be crawled can both be used to deal with crawling for website hosts. Note that this approach does not prevent the data from being redistributed to more vulnerable hosts.

[10]Alternatively, more sophisticated tricks can be implemented to relax this issue, such as decrypting the data on mouse hover, or requiring the decryption key on every viewing, but we leave such investigations to others.

[11]We leave details on the implementation of such a system to the individual institutions. In case of conflicting business incentives, they may restrict exclusion controls to specific users approved for research or find other solutions.

[12]E.g., Wikipedia, PubMed, Twitter, Reddit, and so on.

2015; coreference resolution with Wikipedia entity links, Ghaddar and Langlais, 2016; news article summaries, Fabbri et al., 2019; emojis, Felbo et al., 2017) or manually labeled.

When the data is used for evaluation, it is presented without the context in which the data originally appeared on the internet. Under Scenario 1, however, this context is potentially known to the model: The model can memorize the context in which a given data instance appeared in, and recall this context when the instance is presented to it for labeling. In such cases, we treat the evaluation data as compromised.

The case of automatic labeling, where the label is directly derived from the context, is a trivial case of data contamination and therefore should be avoided when constructing evaluation benchmarks. However, the case of manual annotation should also be carefully scrutinized: If the original context of the test instance contains information that can be helpful to the model in solving the instance, the model can *use this context to cheat the evaluation*, even if the solution was manually annotated. This is a particularly challenging case of data contamination under Scenario 1, as it is difficult to detect when the connection between the context and the instance label is nuanced.

Strategy 3 calls for two actions: (A) Releasing context information alongside the evaluation data. This context is not intended to be used in the evaluation, but as documentation to enable any evaluation practitioner to execute point B, if they wish to scrutinize the integrity of the data; (B) Detecting and discarding instances where the context, which is not intended to serve as input to the model, is indicative of the solution in some significant way. Such instances can be used for training, but are not suitable for a benchmark. In particular, the practice of collecting automatic labels for evaluation tasks from the internet *is fragile under Scenario 1*, and should only be executed with great caution.

## 5 Discussion and Open Questions

**Documenting existing evaluations for contamination.** Research that collects existing test sets into large-scale benchmarks (e.g., Liang et al., 2022; Srivastava et al., 2022) should be attentive to the issues raised in this paper, and in particular employ Strategy 3. This is crucial for benchmarks that incorporate test sets predating the common practice of internet-crawled training corpora, and are

trivially compromised in Scenario 1 in a way that renders Strategies 1 and 2 irrelevant (e.g., review sentiment, Maas et al., 2011). A thorough assessment of the internet presence of current evaluation sets is needed to check what portion of them can be found explicitly in plain text, or implicitly through the origins of the data.

**Centralized strategies.** In this work, we primarily focus on decentralized strategies that are effective at the level of individual researchers or individual evaluation sets. Centralized strategies would be helpful under the assumption that many or all relevant actors can work together from the outset. Although this assumption is more strict, the potential benefits merit additional research.

**Partially effective strategies.** It may be additionally helpful to develop strategies that are only effective under certain conditions, or only partially effective. In Appendix A we discuss one such strategy of templated test examples. Other examples include maintaining a research database of evaluation data and contamination events. This will make it easier and more convenient for model practitioners to make sure that test data is not included in training, or disclosed to this database if the training of a particular model included a certain test example. Such a database is not a strictly effective mitigation strategy, but it may potentially help if model developers cooperate. Another example is *watermarking* data so that it can be detected more easily (Sadasivan et al., 2023; Kirchenbauer et al., 2023). While this is an active field for model-generated data, it is unexplored for watermarking test data for contamination detection, where the watermarking may be applied to *metadata*, or other non-intrusive parts of the data context.

## 6 Conclusions

This paper serves foremost as a call to action to encrypt all evaluation data uploaded to the internet or otherwise evade automatic crawling with gating mechanisms; to withhold from sending unseen evaluation data to closed API models without exclusion controls; to reassess internet-derived evaluation data; and to enrich the discussion around possible robust strategies under more strict assumptions like adversarial actors or negligent actors. We call on the research community to address the issue of data contamination to the best of our ability. In particular, we should be conscious of who and what can access our test data, and plan accordingly.

## Limitations

**On negligent actors.** Negligence can unavoidably decrease or invalidate the effectiveness of the strategies in all cases. For example, a negligent actor may re-upload a plain-text version of a dataset and make it accessible to crawlers. And in particular, Strategy 2 is vulnerable. Nevertheless, researchers can take additional steps to decrease the likelihood of negligence scenarios, or set guidelines for resolving such cases after the contamination event occurred. We highlight two negligence scenarios here:

*"I accidentally sent test data to a closed API model for the first time."* The most important action to take in this scenario is to publicize the event, and notify the original contact behind the test data. Depending on the quantity of compromised examples, it may be possible to continue using the rest of the data, and regardless, the data remains uncompromised for other models. Contacting the API holder to notify them about the identity of the data may also be helpful. And finally, consider collecting replacements to the data, if this is possible.

*"Someone else, unaware of Strategy 2, sent test data to a closed API model."* This is the most vulnerable weakness of Strategy 2. The API holder is the only actor capable of detecting the event, if they have knowledge of what evaluation data to exclude from future training. Otherwise, this scenario may be reduced by adding sufficient warnings to the data's host web-page, or the data file itself. Alternatively, the access to expensive test data can be restricted only to trusted institutions and individuals, although this kind of mechanism may be too restrictive.

**On derivatives.** What constitutes derivative work, or an adaptation, in a legal setting depends on local copyright law, and decryption may not necessarily constitute an adaptation. The purpose of the license in Strategy 1 is not to guarantee the ability to legally enforce encrypted distribution but to encourage it among sincere actors.

Test data licenses that specifically address data contamination scenarios can help in making the strategies more reliable and enforceable. For example, a dataset license that forbids sending its test data to closed API models that don't comply with specific exclusion controls can provide legal means to enforce Strategy 2 downstream; and a license that specifically forbids using the data for model training or requires encrypted distribution can still permit the distribution of derivatives while serving a similar function to a "no derivatives" license.

**On sincere actors.** While the "sincere actors" assumption is generally feasible, naturally it does not always hold. We can conceive one plausible case where it does not hold: When using *crowdsourcing* for large manual annotation efforts, it is certainly possible for some crowdworkers to be adversarial—and use closed API models, such as the freely-available ChatGPT, to automate the labeling against guidelines, if they believe they will not be caught. This will compromise the data, particularly in the case of ChatGPT's web interface, since it does not have training exclusion guarantees as of this writing (Footnote 3). This is an important and challenging scenario to be aware of when using crowdsourcing to annotate test data.

## Acknowledgements

We are grateful to Omar Sanseviero for his helpful comments. This project received funding from the European Research Council (ERC) under the European Union's Horizon 2020 research and innovation programme, grant agreement No. 802774 (iEXTRACT).

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

## A  *Partial Strategy A*: Templated test instances

*Conditions: Scenario 1 or 2.*

This strategy requires significant effort, is not a complete defense, and does not apply in many cases, but is nevertheless one of the few practical strategies to guard against Scenario 2.

For some tasks, particularly with textual data, it is possible to counterfactually augment test data by converting it into a programmatical template.[13] For example, in tasks that require arithmetic reasoning, the answer can change as conditioned on a number, and this can be derived automatically. In summarization tasks, any information in the input (e.g., entity names, dates, or more complex semantic information) should be reflected in the summary.

In test time, for a given evaluation attempt, only one counterfactual variant is sampled with a seed value, and the variant is used across all evaluated systems in that attempt. The seed is forfeit once the evaluation is completed, and the seed (or test data itself) can be publicized.

The more elaborate the counterfactual variant is, the stronger its function as an unseen test case, even if the source instance is compromised. Although this strategy is expensive to implement, it is worth noting that test sets are traditionally small.

---

[13]As discussed by Ribeiro et al. (2020), although in the context unit tests for text models, which are deliberately simple.