# OpenReview forum: "Stop Uploading Test Data in Plain Text: Practical Strategies for Mitigating Data Contamination by Evaluation Benchmarks"
_EMNLP/2023/Conference — EMNLP 2023 Main_

### Official Review · Reviewer_Vde9 · 2023-08-02

**Soundness:** 3

**Excitement:**

3: Ambivalent: It has merits (e.g., it reports state-of-the-art results, the idea is nice), but there are key weaknesses (e.g., it describes incremental work), and it can significantly benefit from another round of revision. However, I won't object to accepting it if my co-reviewers champion it.

**Paper Topic And Main Contributions:**

This work focuses on addressing the threat of data leakage and contamination in closed-source task data caused by the widespread use of text obtained through general web crawling for model training. The authors envisioned specific scenarios and proposed three different countermeasures.

**Questions For The Authors:**

Q1 - How to define the questions and answers within the text, in other words, how to define the test set and training set? In an increasingly free and open language model utilization scenario, text-based answers and questions may become more flexible, and users may inquire from different perspectives. Therefore, each element has the potential to appear in either the question or the answer.

Q2 - The encryption method for text may bring certain inconveniences to the use of normal text. Is it possible to find a better balance between text privacy and convenience of use? Considering question (1), if there is no way to define the boundaries between questions and answers, should all text be encrypted to prevent leakage? Would this significantly disrupt normal communication among people?

**Reasons To Accept:**

The problem addressed in this work is becoming increasingly important with the development of large-scale model technologies.

**Reasons To Reject:**

1 - This work lacks a reasonable task abstraction or problem definition, as well as the absence of mathematical expressions and model training.

2 - There is no theoretical or practical validation to support the authors' viewpoints.



**Reproducibility:**

4: Could mostly reproduce the results, but there may be some variation because of sample variance or minor variations in their interpretation of the protocol or method.

**Reviewer Confidence:**

4: Quite sure. I tried to check the important points carefully. It's unlikely, though conceivable, that I missed something that should affect my ratings.

---

> ### Author Rebuttal · Authors · 2023-08-27
>
> We thank the reviewer for the thoughtful and very detailed review, and for the constructive suggestions! We address the comments below, and will also update the paper with all of the information here for the camera-ready version.
>
> The review mentions several points regarding theoretical formalization, well-defined properties of train and test data, task abstraction, and so on. On this issue, we adhere to simple formalizations inherited from standard machine learning literature (including generative literature), rather than attempt to reinvent the wheel ourselves. We will add and cite such propositions in the paper in the camera-ready version. Notably, our goal here is to discuss real-world grounded and practical manifestations of the theory, rather than the theory itself, which is evidently too general to be plausibly solved in a decentralized and democratized community.
>
> On theoretical or practical validation for our viewpoints (point 2), we would sincerely appreciate it if you could specify which viewpoints fail under scrutiny - to our knowledge, the paper discusses the relevant strengths and assumptions of the strategies, so we could not identify the criticism here.
>
> Regarding the questions:
>
> Q1. Regarding “input” and “output” categorization (test inputs and test outputs), although interesting, it is less relevant for our setting. We advocate for encryption of the entire test set in order to protect it from training, both input and output, since training on test inputs has been documented to also influence performance, and we are concerned with improving the reliability of all evaluations.
>
> Q2. Would encryption of the entire test set disrupt the convenience of current dataset sharing? We argue that this is not the case. In the case of zip-files, though we regard them as convenient, it is also possible to share a small set of compromised unencrypted examples to make showcasing the data more convenient. Any other method of evading web crawlers is also applicable, such as gating mechanisms, e.g., HuggingFace’s “gated repositories” [1], or captcha-like gating mechanisms, and it was not our intent to neglect these solutions.
>
> We will add these discussions to the paper as well, with all relevant details.
>
> Thanks!
>
> [1] https://huggingface.co/docs/hub/datasets-gated

---

### Official Review · Reviewer_Hv9v · 2023-08-04

**Typos Grammar Style And Presentation Improvements:** None
**Soundness:** 5

**Excitement:**

5: Transformative: This paper is likely to change its subfield or computational linguistics broadly. It should be considered for a best paper award. This paper changes the current understanding of some phenomenon, shows a widely held practice to be erroneous in someway, enables a promising direction of research for a (broad or narrow) topic, or creates an exciting new technique.

**Missing References:**

None

**Paper Topic And Main Contributions:**

Addresses a very relevant and pertinent issue in the contemporary world, i.e how Large Language Models trained on internet crawled data could be knowingly or unknowingly be learning from/familiar with data that is used for downstream testing/evaluation.

**Questions For The Authors:**

Would you want to add a line or two about:

- How reliance on such Large Language Models, especially closed ones, is driving the research towards relying on the resources provided by a few big players (only who have access/ability to the expensive training) - not to mention the reliability of black box APIs if the LLM’s model and/or data is closed.

-   How do we know the LLM model is really “learning” as opposed to being a huge memory bank which has memorized subtle patterns, statistical or semantic/especially in context of exposed test data.

- There is also a thin line between reproducibility and data contamination. For example most of the major conferences in NLP demand reproducibility guarantees, which kind of comes with the need for making the test data public. While your encryption key idea might work, how can we ensure that another independent actor who for purely evaluation purposes knowingly or unknowingly uploaded the test data to the internet, and even worse: has no memory of doing so - maybe can add this to the negligent actor paragraph . Maybe one solution there is also uploading a unique SHA key along with the test data. If that key is found say couple of months later in an internet search algorithm, we now know the data along with the key also made its way to the internet. (e.g., OPENAI key found on github is already being flagged). However that will lead to digital verifiability issues- and hence has traces of deep cybersecurity research topics. Maybe that is a collaboration opportunity with the cyber security experts for the next version of this paper.

- Encryption and decryption nevertheless brings in a layer of overhead. Why not a solution based on obfuscation, similar to the ones used by Author Anonymization SOTA. Just making data un-crawlable to machines, while useable with minimal modification to humans

- Or even an extremely unpopular opinion: should results evaluated based on Blackbox models/API/Datasets be considered in a separate track/workshop in NLP conferences. I agree, sad, but it is possibly high time that NLP research community needs to start asking these questions.



**Reasons To Accept:**

- Since large language models, many behind closed API and training datasets are the state of the art now, this paper raises a very time relevant and very important topic to discuss in a big conference like EMNLP

- This paper not only raises these, issues, does a review of existing solutions, analyzes them and shows that they are not sufficient to address the issue of data contamination, but also proposes a few novel solutions.

- This paper is very easy to read and is very well written


**Reasons To Reject:**

None

**Reproducibility:**

N/A: Doesn't apply, since the paper does not include empirical results.

**Reviewer Confidence:**

4: Quite sure. I tried to check the important points carefully. It's unlikely, though conceivable, that I missed something that should affect my ratings.

---

> ### Author Rebuttal · Authors · 2023-08-27
>
> We thank the reviewer for the very uplifting review!
>
> The review mentions various interesting points in the “questions” section. We will certainly add such discussions to the paper in the extra allotted space for camera-ready (within some limits). We also discuss the points below (though we focus on the points with direct implications to the camera-ready version):
>
> The first point regarding community trends and implications on reliability were previously partially discussed in a blog-post [1] which we cite in the paper.
>
> The point regarding reproducibility guarantees is interesting and indeed clashes with negligent/adversarial actor requirements, although perhaps circumventable with limiting access to verified institutions, if such strict regulations are deemed necessary (and this is already the case for some datasets, with its own set of disadvantages). Thank you for the suggestion to the negligent actor discussion, which we will add.
>
> On the point of encryption overhead: Perhaps arguably, we regard password-protected zip files as a convenient and reliable obfuscation format, which is why we put it forward here. Obfuscation can be regarded as encryption without password protection. Unfortunately, some crawlers do implement standardized deobfuscation methods, and implementing a custom obfuscation algorithm is itself inconvenient. Nevertheless, it is true that any method of evading crawlers is valid here - such as captcha gating, HuggingFace gating [2] (though it was not created for this purpose), and so on, and it was not our intent to neglect these solutions. The revised paper will include this information.
>
> Thanks!
>
> [1] https://hackingsemantics.xyz/2023/closed-baselines/
>
> [2] https://huggingface.co/docs/hub/datasets-gated

---

### Official Review · Reviewer_AUDd · 2023-08-05

**Soundness:** 4

**Excitement:**

4: Strong: This paper deepens the understanding of some phenomenon or lowers the barriers to an existing research direction.

**Paper Topic And Main Contributions:**

This paper discusses the problem of data contamination when it comes to model evaluation.
Specifically, the authors consider scenarios where:
- the model is trained on data crawled from the web, and is therefore possible but unfeasible to check for data contamination (i.e., the presence of the test data in the training data)
- we have no information about the data used for model training (e.g., OpenAI's models).

The authors suggest to:
- not release test data without encryption - public encryption is fine if considering a scenario where all users have good intention, and selecting a license that does not allow data redistribution in a different format.
- not evaluate models through APIs on test data if there is no explicit training exclusion option.

**Reasons To Accept:**

- This paper brings the attention of the community on an important issue, as we have no details about the training data of popular LLMs and this compromises our capability to evaluate the performance of such models.

**Reasons To Reject:**

This paper is mainly a call for action - although the suggested solutions are not groundbreaking, I would like to see the paper at the conference. So no reasons to reject.

**Reproducibility:**

N/A: Doesn't apply, since the paper does not include empirical results.

**Reviewer Confidence:**

3: Pretty sure, but there's a chance I missed something. Although I have a good feel for this area in general, I did not carefully check the paper's details, e.g., the math, experimental design, or novelty.

---

> ### Author Rebuttal · Authors · 2023-08-27
>
> We thank the reviewer for the thoughtful and positive review! We appreciate the reviewer’s affirmation that you would like to see the paper at the conference with no reasons to reject, and recognizing the paper as a call for action.

---

### Meta-Review · Area_Chair_NhY1 · 2023-09-04

**Recommendation:** Sound and Exciting
**Best Paper Recommendation:** Yes

**Metareview:**

There was a consensus among the reviewers that this position paper outlines an important problem (models training on publicly available evaluation sets) and proposes a viable mitigation strategy. It would be a clear benefit to the EMNLP conference program and is a clear accept.

**Meta-Review:**

There was a consensus among the reviewers that this position paper outlines an important problem (models training on publicly available evaluation sets) and proposes a viable mitigation strategy. It would be a clear benefit to the EMNLP conference program and is a clear accept.

---

### Decision · Program_Chairs · 2023-10-07

**Decision:**

Accept-Main

**Comment:**

There was a consensus among the reviewers that this position paper outlines an important problem (models training on publicly available evaluation sets) and proposes a viable mitigation strategy. It would be a clear benefit to the EMNLP conference program and is a clear accept.